# Design, Synthesis, Antifungal Evaluation, Structure–Activity Relationship (SAR) Study, and Molecular Docking of Novel Spirotryprostatin A Derivatives

**DOI:** 10.3390/molecules29040864

**Published:** 2024-02-15

**Authors:** Yang-Min Ma, Xia Miao, Bin Jia, Zhao-Yang Sun, Si-Yue Ma, Cong Yan

**Affiliations:** Key Laboratory of Chemical Additives for China National Light Industry, College of Chemistry and Chemical Engineering, Shaanxi University of Science and Technology, Xi’an 710021, China; 200811042@sust.edu.cn (X.M.); jiabin@sust.edu.cn (B.J.); 210811009@sust.edu.cn (Z.-Y.S.); masiyue@sust.edu.cn (S.-Y.M.); 230812183@sust.edu.cn (C.Y.)

**Keywords:** synthesis, spirotryprostatin A derivatives, phytopathogenic fungi, chiral fungicides, succinate dehydrogenase

## Abstract

Phytopathogenic fungi cause plant diseases and economic losses in agriculture. To efficiently control plant pathogen infections, a total of 19 spirotryprostatin A derivatives and 26 spirooxindole derivatives were designed, synthesized, and tested for their antifungal activity against ten plant pathogens. Additionally, the intermediates of spirooxindole derivatives were investigated, including proposing a mechanism for diastereoselectivity and performing amplification experiments. The bioassay results demonstrated that spirotryprostatin A derivatives possess good and broad-spectrum antifungal activities. Compound **4d** exhibited excellent antifungal activity in vitro, equal to or higher than the positive control ketoconazole, against *Helminthosporium maydis*, *Trichothecium roseum*, *Botrytis cinerea*, *Colletotrichum gloeosporioides*, *Fusarium graminearum*, *Alternaria brassicae*, *Alternaria alternate,* and *Fusarium solan* (MICs: 8–32 µg/mL). Compound **4k** also displayed remarkable antifungal activity against eight other phytopathogenic fungi, including *Fusarium oxysporium* f. sp. *niveum* and *Mycosphaerella melonis* (MICs: 8–32 µg/mL). The preliminary structure–activity relationships (SARs) were further discussed. Moreover, molecular docking studies revealed that spirotryprostatin A derivatives anchored in the binding site of succinate dehydrogenase (SDH). Therefore, these compounds showed potential as natural compound-based chiral fungicides and hold promise as candidates for further enhancements in terms of structure and properties.

## 1. Introduction

According to the literature, plant diseases that are caused by phytopathogenic fungi pose a serious threat to crop production and even food safety. One report indicates that crop-destroying fungi contribute to approximately 20% of global crop yield losses, with an additional 10% loss occurring postharvest [1]. Chemical fungicides are traditionally used to control these phytopathogenic fungi. However, the improper use of fungicides can lead to disease resistance and an ecological burden [2]. Consequently, developing novel antifungal agrochemicals that are harmless to humans or animals and safe for the ecological environment is an urgent demand.

Natural products and their derivatives have been extensively studied and utilized for their antimicrobial properties. Spirooxindole alkaloids, which are widely found in natural products (e.g., **1**–**4**; Figure 1), in particular, have shown significant inhibitory effects against various bacteria and fungi [3,4,5,6]. These compounds have a wide range of biological activities, including antimicrobial [7], antiviral [8], and insecticidal activities [9]. Recent studies have demonstrated that synthetic spirooxindole derivatives exhibit high fungicidal activity against phytopathogenic fungi. A series of synthetic spirooxindole derivatives possessed good activities against tobacco mosaic virus (TMV), among which compound **5** and **6** (Figure 1) showed high fungicidal activity levels against *Physalospora piricola*, *Sclerotinia sclerotiorum*, and *Rhizoctonia solani* [8]. Another more recent study showed that spirooxindole analogues had broad-spectrum fungicidal activities against 14 kinds of phytopathogenic fungi and selective fungicidal activities [10]. Among them, compound **7** (Figure 1) is the best of all. It showed the highest antiviral activity in vitro as well as inactivation, curative, and protection activities in vivo, even surpassing ribavirin. Therefore, spirooxindole derivatives have the potential to be developed as effective fungicides, especially for pathogenic pathogens.

Spirotryprostatin A (Figure 1), which has a 3,3′-pyrrolidinyl-spirooxindole motif and is isolated from the fungus *Aspergillus fumigatus* BM939, has been found to inhibit the progression of the mammalian tsFT210 cell cycle in the G2/M phase at micromolar concentrations [11]. In our previous research, spirotryprostatin A derivatives **8** and **9** (Figure 1) showed broad-spectrum fungicidal activity against various fungi, including *Colletotrichum gloeosporioides*, *Valsa mali*, *Alternaria alternata*, and *Alternaria brassicae*, with minimum inhibitory concentrations ranging from 1.1 to 36.9 μM [12]. These derivatives exhibited better activities than their corresponding intermediate spirooxindole derivatives. Therefore, developing efficient and practical methodologies for synthesizing spirotryprostatin A scaffolds and observing their fungicidal activity are significant objectives. It is worth noting that our group was the first to discover the fungicidal activity of spirotryprostatin A derivatives [13]. However, the limited number of available compounds for managing fungal diseases means that developing effective and safe fungicides is a complex process that requires extensive research. Efforts are being made to synthetize novel spirotryprostatin A scaffolds and test their biological activities as antifungals.

It is well established that chirality plays a crucial role in pharmaceuticals, biological molecules, and agrochemicals, with many active ingredients containing asymmetric centers within their structures [14]. In the case of agrochemicals, approximately 30% of active ingredients in modern formulations, including insecticides, fungicides, and herbicides, possess stereogenic centers [15]. Therefore, the development of chiral antifungal agrochemicals is important for effectively inhibiting plant pathogenic fungi [16]. In this context, spirotryprostatin A scaffolds hold great potential as novel chiral fungicide agents targeting plant pathogens. However, the unique structures of chiral spiro-heterocyclic scaffolds pose challenges in their synthesis. While various synthetic methods have been employed for the synthesis of spirooxindole derivatives, such as [3 + 2] cycloaddition reactions [17], Michael addition/cyclization cascades, Heck reactions, and other domino condensations [18,19,20,21], there are only seven reported total syntheses of spirotryprostatin A [22,23,24,25,26,27,28,29], and the synthesis of spirotryprostatin A derivatives has rarely been documented. Moreover, research on the antifungal activity of spirooxindole alkaloids, especially spirotryprostatin A derivatives, is scarce. Therefore, the development of an efficient and highly enantioselective method for synthesizing spirotryprostatin A derivatives would contribute to expanding their structural diversity and exploring their antifungal activities further.

In agrochemical research, succinate dehydrogenase (SDH) has been identified as a significant target for various fungicides [30]. Among them, succinate dehydrogenase inhibitors (SDHIs) have gained considerable attention and emerged as a rapidly growing class of fungicides due to their unique mechanism of action and broad-spectrum inhibitory activity [31]. Many SDHIs contain substituents like -F and -CF_3_, such as bixafen, thifluzamide, fluindapyr, penthiopyrad, and so on [32]. Additionally, the amide boLinend is commonly found in SDHIs. Based on this knowledge, novel spirotryprostatin A derivatives were designed with rich amide bonds and -F and -CF_3_ substituents, hypothesizing their potential as SDHIs (Figure 2). Molecular modeling was also utilized to explore the possible bonding mechanism between these compounds and the target enzyme SDH. This information can guide further experimental design and optimization of the compounds to enhance their activity.

Based on the reasons mentioned above, this study aims to improve the synthetic process of spirotryprostatin A derivatives through transition metal-catalyzed [3 + 2] cycloaddition (Figure 2). This improvement is expected to have significant industrial value due to a high yield, moderate reaction conditions, and a more affordable catalyst. The Ag(I)/(*S*)-Monophos-catalyzed [3 + 2] cycloaddition synthesis was introduced for the preparation of spirotryprostatin A derivatives. Scaling up the reaction under optimal conditions resulted in good yields of spirooxindole derivatives (81–93%) from the template substrates. The fungicidal activities of all the compounds were then tested against ten phytopathogens in vitro, and the preliminary structure–activity relationship was discussed. Additionally, a computational docking study was conducted between active compounds and SDH (according to references [32,33]) to reveal possible antifungal behavior, and enantioenriched spirotryprostatin A derivatives were identified as promising chiral antifungal candidates.

## 2. Results and Discussion

### 2.1. Synthetic Process of Spirooxindole Intermediates ***3***

#### 2.1.1. Optimization of Asymmetric Reaction Conditions

The screening of reaction parameters for the methy-(*E*)-2-((4-fluorobenzylidene) amino) acetate (**1a**) and (*E*)-3-benzylideneindolin-2-one (**2a**) was initiated using a chiral ligand, transition metal Ag(I)/Cu(II), and base (Figure 1 and Table 1). The use of a commercially available (*S*)-Monophos catalyst (**L4**) and AgOAc resulted in the production of spirooxindole derivative **3a** as the main diastereomer. Compound **3a** could be easily separated on silica and isolated with a good yield of 97% and high diastereoselectivity (>20:1, Table 1, entry 5).

Further evaluation of the effect of the transition metal, solvent, and base on the reaction efficiency and stereochemical outcome was conducted (Table 1, entries 9, 10, 11, and 12). It was observed that conducting the model reaction in THF and Et_3_N (Table 1, entry 5) led to a high yield and good stereochemical outcome of spirooxindole **3a**. Therefore, the overall optimal conditions for the reaction were determined as follows: **1** (1.2 eq.), **2** (1.0 eq.), **L4** (5% mol), AgOAc (5% mol), Et_3_N (15% mol), and THF (3 mL), at 0 °C for 8 h.

#### 2.1.2. Scope of Substrates

After optimizing the reaction conditions, the exploration of the scope of the [3 + 2] cycloaddition was initiated by varying the groups of N-arylmethylene glycine methyl ester **1** and 3-benzylidene-2-indolone **2** (Table 2). The investigation revealed that alkyl, halogen, and nitro groups were compatible, and the corresponding intermediates with high diastereoselectivity (**3b**, up to 88:1) were obtained in moderate-to-excellent yields (52–99%). Interestingly, even after introducing a strong electron-withdrawing group, namely, CF_3_ (**3l**, **3m**), the reaction proceeded successfully, resulting in excellent yields (82–99%) and good diastereoselectivity (22:1 and 36:1, respectively). This indicated that the reaction was tolerant of such groups. Additionally, the compatibility of conjugated molecules was examined, and the introduction of the naphthalene ring (**3t**) also led to the formation of corresponding products in good yields (79%) and with good diastereoselectivity (22:1). However, we observed that when the benzene’s substituent of azomethine ylide group **1** was in the *ortho*-position, the reaction could not be efficiently carried out due to the spatial site-blocking effect. This resulted in a significant decrease in the yield and diastereoselectivity of **3k** (55% and 4:1, respectively). Furthermore, we found that the yields of *para*-substituents were higher than those of neighboring substituents, as seen in compounds **3d**, **3g**, **3m**, and **3p** (82–99%). We also investigated the applicability of pro-dipole substrate **2** and found that the introduction of halogen (product **3u**), electron donor substituents (product **3v**), and aliphatic rings (product **3w**–**z**) resulted in corresponding products in moderate-to-excellent yields (65–93%) and good diastereoselectivity (11:1 to 27:1). In total, all of these substrates, **1** and **2,** exhibited good reactivity, affording the corresponding products **3a**–**z** in yields ranging from 55% to 99% with diastereomeric ratios of 4:1 to 74:1. Notably, compounds **3b**–**c**, **3e**, **3h**, **3i**, **3k**, **3l**, **3n**–**r**, and **3t**–**z** were synthesized for the first time. Overall, these findings provide important insights into the substrate scope and the impact of different substituents on the control of stereosynthesis, enhancing the understanding of this asymmetric [3 + 2] cycloaddition for the construction of chiral spirocycles.

#### 2.1.3. Competition Experiment

To further investigate the electronic effect of dipole fragments on the reaction, a competition experiment was designed using substrates **1m** and **1s** (Figure 2). Both substrates were added simultaneously under the optimal reaction conditions, and the presence of the two compounds was monitored after the completion of the reaction. The results of the competition experiment revealed that the conversion of **1m** reached 82%, while only 9% of the **1s** counterparts underwent the reaction to form the corresponding compounds. This indicates a higher reactivity of dipoles with electron-withdrawing groups, emphasizing the electronic effect of the dipole fragments on the reaction. The observation that the reaction favored electron-withdrawing groups further supports our previous findings on the positive role of such groups in regulating the stability and reactivity of the reaction. This information is valuable for designing and optimizing future reactions that are similar dipole fragments and provides insights into the electronic factors governing reactivity in these types of transformations.

#### 2.1.4. Amplification Experiments

The progress that was made in scaling up the reaction is impressive, especially in the context of improved reactivity for the electron-donating group-substituted dipole and the successful reaction with the strong electron-withdrawing group-substituted dipole (Figure 3). Template substrates **1a** and **2a** could still provide a good yield, and compound **3a** was obtained in an 84% yield. It was surprising to find that the reactivity of the electron-donating group-substituted dipole **1s** was improved after scaling up the reaction, and the yield reached 93%, while the strong electron-withdrawing group-substituted dipole **1m** was able to react with **2a** smoothly, and compound **3m** was still obtained in a moderate yield of 51%. The potential application of this developed method was exemplified by the synthesis of products **3a** and **3s** on a larger scale and their subsequent diverse functional transformations, as shown in Figure 3 and Table 2. Hence, the developed synthetic protocol is well suited for scaling up the reaction in the late stage, which would facilitate industrial production.

#### 2.1.5. Analysis of Stereo Configuration of Spirooxindole Intermediate **3**

Based on the absolute configuration of products and previous reports [34,35], a transition state was proposed to rationalize the stereochemical outcome of the process, shown in Figure 3A. The chiral ligand, (*S*)-MonoPhos, initially forms a chiral complex with Ag(I), which then coordinates with the azomethine ylide 1 to selectively form a tetrahedral complex. Subsequently, 2-oxoindolin-3-ylidene 2 attacks from the less hindered side. Due to steric hindrance, the aromatic substituent group of the 2-oxoindolin-3-ylidene 2 is positioned closer to the -CO2Me moiety of the azomethine ylide 1. It is possible that the oxygen atom of (*S*)-MonoPhos and the carbonyl group of the indolone experience repulsion, resulting in a twist towards the direction of the carbonyl group. This twist leads to the formation of the adduct in the (2′R,3S,4′R,5′R) configuration [35,36], which is consistent with the analysis from X-ray crystallography, shown in Figure 3B,C [37]. However, further investigation is still needed to elucidate the actual catalytic mechanism.

### 2.2. Structures of Target Compounds

The intermediate **3a** was used to synthesize spirotryprostatin A derivatives **4a** via the classical Schotten–Baumann reaction with Fmoc-L-pro-Cl, followed by deprotection and cyclization, as depicted in Figure 4. The target products were obtained in moderate-to-good yields (41–77%), as shown in Figure 4, with compounds **4b***–***g**, **4i**, and **4k***–***r** as the first synthesized compounds. The additional chiral center in the target products, derived from the L-proline starting material, allows for the determination of the stereoconfiguration based on previous studies [15]. Despite the moderate yields, these spirotryprostatin A derivatives represent an important milestone in the development of this method. Further exploration of reaction conditions and substrate modifications may lead to improved yields and selectivity in the future.

### 2.3. Antifungal Activity In Vitro

To improve the fungicidal activities in vitro of the target compounds, 19 spirotryprostatin A derivatives (**4a***–***4s**) and their corresponding synthetic spirocyclic intermediates (**3a***–***3s**) were screened using the filter paper sheet method. Antifungal activities were evaluated at 1 mg/mL, with ketoconazole as the positive control, against 10 phytopathogens. The majority of the title compounds exhibited a broad spectrum of biological activities against various phytopathogens, with some showing superior inhibitory effects compared to the positive control. Notably, compounds **4d**, **4e**, **4g,** and **4h** exhibited strong antifungal activities against H. maydis, while compounds **3d** and **3k** displayed strong activities against A. brassicae. Compound **4d** also demonstrated remarkable activity against M. melonis. Additionally, nearly half of the title compounds exhibited good activities compared to the positive control. Overall, the target compounds **4a***–***4s** generally showed higher inhibitory activities than the spirocyclic intermediates **3a***–***3s**, but both groups demonstrated significant inhibition against the 10 pathogenic fungi that were tested. Consequently, further experiments were conducted to analyze their antifungal activities accurately, utilizing the equipartition dilution method.

The MIC values of the compounds (**3a**–**3s** and **4a**–**4s**) were determined to accurately evaluate the growth inhibition of 10 pathogenic pathogens and to gain a comprehensive understanding of the inhibitory effects of compounds **3a**–**3s** and **4a**–**4s**. The inhibition data were selectively analyzed, and they can be visualized using Table 3. Obviously, the fungicidal activity data that are shown in Table 3 and Table 4 share some characteristics. Most compounds demonstrated good antifungal activities, with MIC values ranging from 8 to 64 µg/mL. Both classes of compounds, **3a**–**3s** and **4a**–**4s**, exhibited a broad antifungal spectrum and strong inhibitory effect against the tested phytopathogenic fungi, although compounds **4a**–**4s** generally displayed higher activities compared to **3a**–**3s**. Notably, both groups exhibited potent fungicidal activities against *T. roseum* (MICs: 8–32 µg/mL). It is particularly noteworthy that compounds **4a**–**4s** exhibited excellent inhibitory effects on the tested phytopathogenic fungi. This suggests potential applications of spirotryprostatin A derivatives in agriculture and plant protection. Furthermore, compounds **4a**–**4q**, **4s**, and **4r** were found to be the most effective against *T. roseum* and *F. solani* in the range of 8–32 µg/mL, surpassing or equaling the positive control ketoconazole. As shown in Table 4, the derivatives displayed moderate-to-good fungicidal activities against *B. cinerea*, *F. graminearum*, *A. brassicae,* and *A. alternate* (MICs: 8–64 µg/mL). Compounds **4k** and **4m** exhibited lower MIC values than the positive control against *B. cinerea*, and compound **4j** showed a lower MIC value than the positive control against *A. alternate*. However, most of the spirotryprostatin A derivatives displayed a lower antifungal activity range (32–128 µg/mL) against the four test fungi, *H. maydis*, *C. gloeosporioides*, *FON*, and *M. melonis*, compared to ketoconazole. Fortunately, the MICs of compounds **4i**, **4m**, and **4o** were equal to the positive control against *C. gloeosporioides* (8 µg/mL).

It has been proposed that ligands with a Clog P value of less than 5 have a more promising drug-likeness profile [38]. Therefore, the Clog P values of spirotryprostatin A derivatives (**4a**, **4d**, **4k**, **4q**) were calculated, and valuable results (Clog P, 3.24–4.21) were obtained, as shown in Appendix A. These results suggest that the compounds can be potential candidates.

In summary, the study demonstrated that both spirotryprostatin A derivatives (**4a**–**4s**) and their synthetic spirocyclic intermediates (**3a**–**3s**) exhibited a broad range of inhibitory effects against 10 different tested phytopathogens. The spirotryprostatin A derivatives (**4a**–**4s**) generally displayed higher antifungal activities compared to their intermediates (**3a**–**3s**). These findings revealed that spirotryprostatin A derivatives have the potential to serve as new chiral antifungal agents or highly active lead compounds for the development of natural product-derived antifungal agents.

### 2.4. Structure–Activity Relationship

By comparing the initial activities (Table 3) and MIC values (Table 4), the study derived the primary structure–activity relationships. It was found that spirotryprostatin A scaffolds had a more significant impact on antifungal activity compared to spirocyclic skeletons, indicating the crucial role of spirotryprostatin A scaffolds in enhancing antifungal capacity (Figure 5). The study also observed trends in the effects of substituents on the antifungal activity of spirotryprostatin A derivative **4a**. First, the type of substituent had a significant influence, with compounds that were modified with electron-deficient groups (-F, -Cl, -Br, and -CF_3_) (**4b**–**4m**) showing excellent inhibitory activities, while those that were modified with an electron-donating group (-OCH_3_) exhibited moderate inhibitory activities (**4q**–**4s**). This suggests that modifying spirotryprostatin A scaffolds with electron-deficient groups enables us to broadly enhance their bioactivities. Secondly, the position of the substituents on the phenyl ring A was explored. *Meta*-substitutions (**4c**, **4f**, **4i**, **4o**, **4r**) or *para*-substitutions (**4d**, **4g**, **4j**, **4p**, **4s**), regardless of electron-donating or electron-withdrawing groups, exhibited a stronger inhibitory effect on pathogenic fungi compared to neighboring positions (**4b**, **4e**, **4h**, **4n**, **4q**). This indicates that the 3′- and 4′-positions seem to be relatively important active sites. For example, the introduction of -OCH_3_ (**4q**) at the 2′-position, on the other hand, resulted in a weaker antifungal activity compared to 4r and 4s against *A. alternate*, *F. solani*, and so on. The aforementioned research demonstrates that the presence of electron-withdrawing groups at the *meta*- (**4c**, **4f**, **4i**, **4l**, **4o**) and *para* positions (**4d**, **4g**, **4j**, **4m**, **4p**) exerted a favorable influence on biological activity, namely, F, Cl, Br, CF_3_, and NO_2_. Hence, the occurrence of electron-donating groups at positions 3′ and 4′ (**4r**, **4s**), where the Hammett constant is negative, may potentially decrease the biological activity [39]. Moreover, the investigation revealed that with an increase in the Hammett constant, the corresponding biological activity generally exhibited an upward trend [40]. The study also noted that the spirooxindole intermediates with -OCH_3_ introduced in the *para* position (**3s**) were more active than the other positions (**3q**/**3r**). Therefore, unsatisfactory electronic and spatial effects apparently prevented obtaining highly active products. However, when -CF_3_ (**4k**) was introduced at the 2′-position of the phenyl ring A, the compounds exhibited broad-spectrum inhibitory activity and showed particularly significant performance. This feature was more easily adapted to compound **3k** compared to other spirocyclic intermediates. Compounds containing -F and -CF_3_ groups (**4d** and **4m**) displayed more significant and broad-range inhibitory activity, indicating the significant role of the -CF_3_ group in structural modification and regulating physiological activity. 

It is worth mentioning that some of the plant pathogenic fungi have been successfully controlled using SDHIs, including *B. cinerea* [41], *A. alternate* [42], and so on. It was found that compounds **4k** and **4m** exhibited significant inhibitory activities against *B. cinerea* and *A. alternate*, which is a delightful discovery. Furthermore, the inhibitory activity of **4k** and **4m** was found to be even more potent than the positive control ketoconazole against *B. cinerea*. This suggests their potential as fungicides or antifungal agents specifically targeting these pathogens, and further investigation should be conducted. 

### 2.5. Molecular Docking

The binding pattern of compound **4d** to SDH is shown in Figure 6, where it can be seen that compound **4d** is deeply invaginated into the SDH protein, stably bound in an active pocket consisting of the B, C, and D chains of SDH (Figure 6B). For example, **4d** is located in a hydrophobic luminal pocket composed of amino acid residues B/Pro-169, B/Trp-172, B/Trp-173, C/Ile-30, C /Ile-218, and C/Trp-35, forming a strong hydrophobic interaction. A detailed analysis shows that the oxygen atom of the indole ring of **4d** can form hydrogen bonding interactions with amino acid residues D/Try-91 and B/Try-173, respectively, which are the major interactions between compound **4d** and SDH (Figure 7B). All these interactions resulted in the formation of a stable complex between **4d** and SDH; in addition, the molecule showed good binding free energy (ΔGb = −9.36 kJ/mol) toward the target protein. The binding pattern of **4a** with SDH was similar to that of **4d** (Figure 7A). In particular, compound **4k** formed a strong halogen bond with amino acid residue C/Met-39 during the binding process (Figure 7C), which further stabilized the ligand–receptor complexes. The binding free energy of spirooxindole intermediates (**3a**, **3b**, **3d**, **3k**, **3q**) was also evaluated one by one, as can be seen in Appendix A. The intermediates bound to SDH with higher binding free energy (from −7.28 to −7.40 kJ/mol), significantly weaker than spirotryprostatin A derivatives (**4a**, **4d**, **4k**, **4q**), from −8.93 to −9.42 kJ/mol. This suggests that the antifungal activities of these two groups of compounds are seemingly derived from the interaction between the compounds and the enzyme SDH. In summary, it can be hypothesized that the antifungal effect of the target compounds may be achieved by binding with SDH. It is important to note that molecular modeling predictions are based on computational algorithms and assumptions, and they should be validated through experimental studies. Nonetheless, molecular modeling provides valuable insights into the structural aspects of compound–target interactions.

## 3. Materials and Methods

### 3.1. Instruments and Chemicals

All reagents and chemicals used in the experiments were purchased from commercial sources and were used without further purification, as they met the requirements of the experimental procedure. The progress of the reactions was monitored via thin-layer chromatography (TLC) using Silica gel 60 GF254. Melting points were measured on an X-6 micro-melting point apparatus and were uncorrected. Nuclear Magnetic Resonance (NMR) spectra were recorded using a Bruker 400 NMR instrument. CDCl_3_ and DMSO-*d*_6_ were used as solvents for NMR experiments, with tetramethylsilane (TMS) as the internal standard. The single-crystal structure of the target compound was determined using a Bruker D8 Quest (Corporation, Bruker AXS Ltd., Karlsruhe, Germany) diffractometer to obtain accurate structural information. This instrument was used to analyze the crystal structure and obtain accurate structural information of the compound. All of the reactions were conducted under an Ar atmosphere using standard Schlenk techniques. Glassware was dried in an oven (100 °C) and heated under reduced pressure before use. Mass spectra were obtained utilizing a Bruker Impact HD Q-TOF high-resolution mass spectrometer.

### 3.2. General Synthetic Method of Spirooxindole Intermediates (***3a***)

We synthesized azomethine ylide (**1a**) and protic-dipolar (**2a**) compounds based on previous work [13] for subsequent reactions. In a 10 mL Schlenk flask, silver acetate (AgOAc, 4.2 mg, 5% mol) and the chiral ligand (*S*)-Monophos (9.0 mg, 5% mol) were dissolved using tetrahydrofuran (THF, 1 mL) under an argon atmosphere. After stirring at 0 °C for 1 h, azomethine ylide **1a** (0.6 mmol), 2-oxoindolin-3-ylidene **2a** (0.5 mmol), and triethylamine (TEA, 15% mol) were dissolved in THF (2 mL) and injected into the reaction flask under an argon atmosphere. The reaction mixture was maintained at 0 °C and stirred for an additional 8 h. The pure intermediate **3a** was yielded via silica gel column chromatography (petroleum ether/ethyl acetate = 10:3, *v*/*v*).

### 3.3. General Synthetic Method of Target Compounds (***4a***)

Initially, the spirocyclical intermediate **3a** (0.1 mmol) was introduced into a two-phase reaction system consisting of dichloromethane (DCM, 5 mL) and saturated sodium carbonate (Na_2_CO_3_, 5 mL). Subsequently, a solution of Fmoc-L-Pro-Cl (0.12 mmol) was added dropwise at a controlled rate of 2–3 drops per second, followed by a 4 h reaction period at ambient temperature. After the reaction, the organic layer was extracted with DCM and saturated brine, and then dried over anhydrous magnesium sulfate (MgSO_4_). Piperidine (0.12 mmol) was then added to the reaction mixture and allowed to react for an additional hour at room temperature. The pure intermediate **4a** was obtained by subjecting the reaction mixture to silica gel column chromatography using a mixture of petroleum ether/ethyl acetate (1:1, *v*/*v*) as the eluent.

### 3.4. Fungi

Ten strains of phytopathogenic fungi, *Helminthosporium maydis* (*H. maydis*), *Trichothecium roseum* (*H. roseum*), *Botrytis cinerea* (*B. cinerea*), *Colletotrichum gloeosporioides* (*C. gloeosporioides*), *Fusarium graminearum* (*F. graminearum*), *Alternaria brassicae* (*A. brassicae*), *Alternaria alternate* (*A. alternate*), *Fusarium oxysporium* f.sp. *niveum* (*F. maydis*), *Mycosphaerella melonis* (*M. melonis*), and *Fusarium solani* (*F. solani*), were donated by Xi’an Medical College, Shaanxi, China. The above fungi were cultivated on potato dextrose agar (PDA) plates at 28 °C and kept at 4 °C with periodic subculture.

### 3.5. Antifungal Activity Assay In Vitro

The maximum inhibitory concentration (MIC) values represent the lowest concentration of a sample that inhibits the visible growth of a microorganism. The inhibitory activity and MIC values of spirocyclic pyrrolidone intermediates and indolinedione piperazine alkaloid compounds against plant fungi were determined using the filter paper sheet method [43] and two-fold dilution method [44], respectively. 

Fresh PDA medium was prepared and subjected to sterilization at 120 °C for 30 min. After sterilization, the medium was thoroughly shaken and inverted onto a plate. The compounds under investigation were prepared as a drug solution with a concentration of 1 mg/mL. Ketoconazole, at the same concentration, was used as the positive control, while DMSO served as the blank control. Next, 200 μL of the respective bacterial suspension was inoculated onto the culture medium using the plate spreading method. A circular filter paper sheet with a diameter of 6 mm, impregnated with the drug solution, was placed in the center of the medium after 24 h of natural drying. The plates were incubated at a constant temperature of 28 °C for 48 h. After incubation, the diameter of the transparent inhibitory circle was measured using the crosshatch method. To ensure statistical reliability, each treatment was performed in triplicate.

The 96-well plate and the test solution were UV sterilized. Then, 100 μL of freshly sterilized potato dextrose broth (PDB) medium was added to each of the 12 wells in row A. Next, 100 μL of the test solution was pipetted into well A1 and diluted from left to right to A10 using the two-fold dilution method. The concentrations of the compounds were 256 μg/mL in A1, 128 μg/mL in A2, and in decreasing order 64, 32, 16, 8, 4, 2, 1, and 0.5 μg/mL for the following wells, respectively. Well A11 was used as the PDB control and well A12 as the DMSO blank control group. Finally, the corresponding plant fungal suspension (10^6^ CFU/mL) was added to all 12 wells, and the plate was incubated at a constant temperature of 28 °C for 48 h. The MIC values were recorded after observation and were repeated three times for each group. 

The relevant bioactivity experiments were conducted in an ultra-clean sterile console (SW-CJ-1 FD, Purification Equipment Co., Ltd., Shanghai, China). Vertical high temperature steam sterilizer (LDZF-50L, Shanghai Shen’an Medical Equipment Factory, Shanghai, China) and constant temperature oscillation incubator (HZQ-F160A, Shanghai Yiheng Technology Co., Ltd., Shanghai, China) were used for testing.

### 3.6. Molecular Modeling

The structure of succinate dehydrogenase (SDH: PDB 1ZOY) was obtained from the RCSB protein Data Bank (http://www.rcsb.org accessed on 11 February 2024). AutoDock 4.2 was employed to analyze the interactions of the active compounds and the enzyme. All the heteroatoms were removed from the 1ZOY.pdb to make the complex receptor free of any ligand before docking. Water molecules of enzyme were removed, and hydrogen atoms were added in the standard geometry before docking by AutoDock tools. The ligand file was minimized to the lowest energy, and the standard 3D structure was obtained in (.pdb) format following the protocol described previously [26]. Docking runs were carried out using a radius of 40 Å, with coordinates x = 86.459, y = 65.6, and z = 85.537 and a spacing of 0.375. Docking was conducted by Lamarckian Genetic Algorithm (LGA). PyMOL (version 0.99; DeLano Scientific, San Carlos, CA, USA) and ligplus v1.4.5 were used to view the graphic.

## 4. Conclusions

In summary, a series of novel spirooxindole derivatives and spirotryprostatin A derivatives were designed and synthesized using the [3 + 2] cycloaddition and a chiral catalyst, (*S*)-Monophos, in a satisfactory diastereoselective manner. In addition, a possible pathway for this spiroannulation reaction was also suggested. Advantages of this method include readily available starting materials, good yields, and mild reaction conditions, making it suitable for late-stage scale-up reactions, as shown in further amplification experiments. This indicates that the synthetic route may be feasible for industrial production. Furthermore, most of the compounds exhibited good antifungal activities against the ten typical plant pathogenic fungi, especially the spirotryprostatin A derivatives, which showed a broad range and good inhibition effects. Combined with the computational results of the AutoDock molecular docking method, the inhibitory activity of the target alkaloids on the growth of the plant pathogenic fungi and their structural relationships were revealed for the first time. There could be some correlation between the bioactivity and the SDH inhibitory activity of these compounds. The SAR results showed that the spirotryprostatin A skeleton had significant effects on the activity, and the type and position of substituent also had significant influences on the antifungal activity. All of the results revealed that spirotryprostatin A derivatives are potential plant pathogen inhibitors, which could be further optimized and developed as new agricultural fungicides.

## Data Availability

Data are contained within the article and Appendix A.

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
