# Peer review of "Design, Synthesis, Antifungal Evaluation, Structure–Activity Relationship (SAR) Study, and Molecular Docking of Novel Spirotryprostatin A Derivatives"

_molecules, 2024, doi:10.3390/molecules29040864_

Round 1
Reviewer 1 Report
Comments and Suggestions for Authors
In the manuscript entitled “Design, Synthesis, Antifungal Evaluation, Structure-Activity Relationship (SAR) Study, and Molecular Docking of Novel Spirotryprostatin A Derivatives” the author designed, synthesized and tested new spirotryprostatin A derivatives and spirooxindole derivatives for their antifungal activity against ten plant pathogens. Additionally, they combined with the computational results of the AutoDock molecular docking method, the inhibitory activity of the target alkaloids on the growth of the plant pathogenic fungi and their structural relationships were revealed for the first time. Also, from the aspect of SAR studies, it is important that the skeleton of spirotriprostatin A has been found to have significant effects on activity, and the type and position of the substituent also have a significant effect on antifungal activity.
Topic is current, interesting and the study is comprehensive. The manuscript is well written, organized and has science and technological significance.
Given all the above, I suggest acceptance of the manuscript in this form.
Author Response
I would like to express my sincere appreciation for the your encouraging and supportive feedback you have provided on my manuscript titled [Design, Synthesis, Antifungal Evaluation, Structure-Activity Relationship (SAR) Study, and Molecular Docking of Novel Spirotryprostatin A Derivatives]. I am genuinely grateful for your time and expertise, Thank you once again for your invaluable input.
Reviewer 2 Report
Comments and Suggestions for Authors
The key step of all analysed synthesis is the [3+2] cycloaddition reaction. The description of this transformation require however fundamental improvements. In particular:
- Many molecular systemes defined earlier as "1,3-dipole" exhibit not dipolar nature. So, all terms such as "1,3-dipole", "1,3-dipolar cycloaddition" (and similar) should be replaced to "three atom component", "[3+2] cycloaddition" etc. respectively. The source of these-type nomenclature should be cited at this point [Eur. J. Org. Chem. 267–282 (2019)].
- Schemes 1-3: the step of the formation of azomethine ylide must be included.
- The question of the application of [3+2] cycloaddition reactions for the synthesis of five-membered heterocycles (including reactions with the participation o fazomethine ylides) was very recently reviewed [Scientiae Radices, 2, 247 (2023)].
- Figure 3 suggest the one-step mechanism of the reaction. According to actual state of knowledge, this issue can not be assumed apriori without respective mechanistic experiments. At this moment different type mechanisms (polar or non-polar and one step, one step - two stage, stepwise) should be considered regarding to all [3+2] cycloaddition. This explanation should be included with respective references from the recent references.
- Analysed reaction can theoretically realised on two regioisomeric paths. In the practice however, only one is detected. How can explain the observed regioselectivity?
Other comments:
+ Caption of the Figure 5: Hammet constants for EDG and EWG substituents should be specified.
+ Molecular docking: the obtained results should be compared with similar study for other nitrogen-containing 5-membered heterocyclic systems.
These-type data are aveilable in the literature.
Author Response
Thank you for taking the time to review our manuscript titled [Design, Synthesis, Antifungal Evaluation, Structure-Activity Relationship (SAR) Study, and Molecular Docking of Novel Spirotryprostatin A Derivatives]. I appreciate your valuable comments and suggestions, which have helped improve the quality of our work. In this response, we will address each of your points and explain the changes we have made accordingly. We have prepared a point-by-point response, please see the attachment. Once again, I would like to express my gratitude for your insightful comments.

Reviewer 3 Report
Comments and Suggestions for Authors
The manuscript “Design, Synthesis, Antifungal Evaluation, Structure-Activity Relationship (SAR) Study, and Molecular Docking of Novel Spirotryprostatin A Derivatives” is devoted to the synthesis of a series of spirotryprostatin A derivatives and evaluation of their biological activites (in vitro and in silico). The research is comprehensive and is of high practical significance, since the products are of high interest for drug discovery and agriculture.
The authors have performed a good optimization of the earlier reported reaction to develop a synthetic procedure to spirotryprostatin A derivatives.
The main weaknesses of the paper are poor structure elucidation explanations, poor explanations on absolute configuration determination and absence of ee determination. The studied compounds have very difficult structures…
In my opinion, this manuscript suits to the scope of Molecules. I recommend that after major revision (especially comment 4), it can be accepted.
Some more comments for the authors:
1. Line 458: with a single conformation. What do you mean?
2. Fig. 2: a ref to previous work is required.
3. How absolute configurations were determined?
4. What is ee of the reaction?
Author Response

(The authors gave the same response as above.)

Reviewer 4 Report
Comments and Suggestions for Authors
The manuscript entitled "Design, Synthesis, Antifungal Evaluation, Structure-Activity Relationship (SAR) Study, and Molecular Docking of Novel Spirotryprostatin A Derivatives” (Molecules-2856196) by Ma et al. describes the design, synthesis of novel spirotryprostatin A derivatives. They have also studied molecular docking and SAR for these new molecules. However, the manuscript needs attention in terms of the purity of the compounds and the effect of impurities on the antifungal activity.
1. Impurities are present in most of the final compounds. See spectra Fig S74, S89, S92, S95, S133, S145, S148, S151, S166, S169etc. Purity of the compounds interferes with biological activity. Did authors confirm the purity of the compounds and what percent of purity is accepted for plant diseases?
2. Show expanded spectra for Fig S130, Fig S133
3. All the schemes should be refined, chemdraw structures should be cleaned and reaction conditions (0 °C) should be added.
4. References need attention, E.g. Ref 29, Journal name.
Comments on the Quality of English LanguageNone
Author Response

(The authors gave the same response as above.)

Round 2
Reviewer 3 Report
Comments and Suggestions for Authors
The authors provided necessary changes according to my review.
Author Response

(The authors gave the same response as above.)

Reviewer 4 Report
Comments and Suggestions for Authors
Reaction conditions should be added on schemes including time of reactions, I cannot see °C on the manuscript version I received.
Comments on the Quality of English LanguageNone
Author Response

(The authors gave the same response as above.)
